# Simultaneous Quantification of 66 Compounds in Two Tibetan *Codonopsis* Species Reveals Four Chemical Features by Database-Enabled UHPLC-Q-Orbitrap-MS/MS Analysis

**DOI:** 10.3390/molecules29215203

**Published:** 2024-11-03

**Authors:** Zhouli Xu, Rongxin Cai, Hanxiao Chai, Shaoman Chen, Yongbai Liang, Xican Li, Guihua Jiang

**Affiliations:** 1State Key Laboratory of Southwestern Chinese Medicine Resources, School of Pharmacy, Chengdu University of Traditional Chinese Medicine, Chengdu 611137, China; xuzhouli@stu.cdutcm.edu.cn (Z.X.); choi_roy@foxmail.com (R.C.); 2School of Chinese Herbal Medicine, Guangzhou University of Chinese Medicine, Guangzhou 510006, China; 20221110228@stu.gzucm.edu.cn (H.C.); 20221110152@stu.gzucm.edu.cn (S.C.); 20231110153@stu.gzucm.edu.cn (Y.L.)

**Keywords:** *Codonopsis canescens* Nannf., *Codonopsis nervosa* Nannf., UHPLC-Q-Orbitrap-MS/MS, four chemical features, simultaneous quantification, quality marker

## Abstract

*Codonopsis canescens* Nannf. (CoC) and *Codonopsis nervosa* Nannf. (CoN) are two traditional Tibetan medicinal herbs (Zangdangshen), which have been widely used in the treatment of various diseases. In this study, their aerial and underground parts were systematically analyzed using database-enabled UHPLC-Q-Orbitrap-MS/MS technology. This technology introduced three adduct ions, [M − H]^−^, [M + H]^+^, and [M + NH_4_]^+^, to putatively identify a total of 66 compounds. During the putative identification, at least 16 isomers were successfully differentiated, such as isochlorogenic acid A vs. isochlorogenic acid B vs. isochlorogenic acid C. Thereafter, all these identified compounds were further quantified for their contents based on a linear regression method. Their contents were observed to vary from 0.00 to 39,127.03 µg/g. Through multiple comparisons of these quantification results, the study found the following four chemical features: (1) Four sesquiterpenes (especially atractylenolide III) enriched mainly in CoC and rarely in CoN; (2) four quinic acid derivatives were abundant in the aerial part of two species; (3) sixteen flavonoids (particularly diosmetin and chrysoeriol) showed higher content in CoC than in CoN; and (4) lobetyolin was ubiquitously distributed in four parts of both CoC and CoN. Based on these features and the relevant principles, four compounds (lobetyolin, atractylenolide III, diosmetin, and chrysoeriol) are recommended as the quality markers of two Tibetan *Codonopsis* species. All these findings can facilitate the sustainable development and quality control of the two traditional Tibetan medicinal herbs.

## 1. Introduction

Tibetan *Codonopsis* has long been utilized in traditional medicine. It can originate from several species, including *Codonopsis canescens* Nannf. (CoC) and *Codonopsis nervosa* Nannf. (CoN, Figure 1). CoC and CoN are mainly distributed in the Tibet Autonomous Region and the provinces of Sichuan, Qinghai, Yunnan, and Gansu in China [1]. They exhibit various traditional efficacies, such as reducing inflammation, expelling wind and dampness, as well as strengthening the spleen and stomach. In clinical practice, these two species, among other species, have been extensively employed in the treatment of rheumatoid arthritis, leprosy, neural paralysis, and spleen–stomach deficiency [1,2,3]. Due to the depletion of other wild species, the cultivation and use of CoC and CoN have become a mainstream issue.

Until now, at least four research teams have undertaken efforts to analyze the bioactive compounds in CoC and CoN. In 2009, Fan’s team performed a qualitative analysis of 36 compounds from the aerial parts of CoN [4]. In 2010, Peng’s team identified six compounds from the whole herb of CoN [5]. In 2012, Er-Bu and colleagues identified 15 compounds from the aerial parts of CoN [6]. The most recent study, conducted in 2022 by Wang’s team, analyzed 34 compounds from the aerial parts of CoC [7]. However, these studies encountered scientific challenges which limit the reliability of their conclusions. For instance, Fan’s study did not employ authentic standards to fulfill the accurate compound identification and thus failed to effectively differentiate 15 isomers, such as α-eudesmol, β-eudesmol, and γ-eudesmol [4]. Similarly, Wang’s research lacked authentic standards and thus did not differentiate four isomers, including astragalin and luteolin-7-O-glucoside [7]. As a result, all these previous findings have been classified as “tentative identifications”.

At present, the natural resources of CoC and CoN are gradually depleting in the Tibetan region. On the other hand, a poor understanding of the chemistry has prevented the quality evaluation for both CoC and CoN. As a result, there have been increasing adulterants and substitutes in the medicinal herb market [8].

To address these issues, the current study has collected sufficient authentic standards to establish a database to enable UHPLC-Q-Orbitrap-MS/MS analysis. By means of database-enabled UHPLC-Q-Orbitrap-MS/MS analysis, the study attempted to putatively identify various compounds in CoC and CoN [9,10]. Based on these results, all identified compounds were further quantified using calibration curves and a linear regression of the authentic standards. All these qualitative and quantitative results were further compared between CoC and CoN as well as between the aerial part and underground part. These multiple comparisons provide a systematic and comprehensive chemical understanding of the two Tibetan *Codonopsis* species.

## 2. Results and Discussion

### 2.1. Qualitative Analysis by Database-Enabled UHPLC-Q-Orbitrap-MS/MS

Compared with traditional detection methods such as thin-layer chromatography (TLC) and HPLC, UHPLC-Q-Orbitrap-MS/MS, as an advanced analytical method, was able to provide more comprehensive characterization [11]. In this study, the UHPLC-Q-Orbitrap-MS/MS method was enabled by a database obtained through the collection of authentic standards. The database-enabled UHPLC-Q-Orbitrap-MS/MS analysis was applied for the putative identification of compounds in CoC and CoN. As shown in Figure 2, a total of 66 compounds were identified in the study, including sesquiterpenes, quinic acid derivatives, flavonoids, and other categories. These putative identifications were performed using the following three adduct models: [M − H]^−^, [M + H]^+^, and [M + NH_4_]^+^. Among these adducts, [M − H]^−^ and [M + H]^+^ have been commonly used in the previous UHPLC-Q-Orbitrap-MS/MS analyses [11]. However, the identification of lobetyolin was performed using [M + NH_4_]^+^. This emphasized that different compounds might be influenced by distinct adduct models in mass spectrometry. The corresponding retention time (R.T.), molecular ion, exact mass, experimental mass, mass error and fragments are listed in Table 1.

Based on this analysis, the MS/MS spectra of the compounds were fully elucidated and are presented in Appendix A. These elucidations indicate that distinct structural frameworks may follow different fragmentation pathways. For example, the flavonoid skeletons tend to follow *retro*-Diels-Alder (RDA) fragmentation pathways to open their C-ring (e.g., *m*/*z* 133 and 151 in luteolin **47**, Appendix A).

Notably, through putative identification, 16 isomers were successfully differentiated from each other as well. For example, three dicaffeoylquinic acid isomers, isochlorogenic acid B (**34**), isochlorogenic acid C (**40**), and isochlorogenic acid A (**43**), were differentiated from each other (Figure 3). In the comprehensive analysis of compound **34**, the δ values for *m*/*z* 515 ([M − H]), 353, 191, 179, 135, and 93 were calculated between the theoretical and experimental values. The calculation revealed that the *m*/*z* 353 fragment exhibited an extremely low error between the calculated and experimental values, with a δ value of −1.13 ppm [(353.0873 − 353.0877)/353.0873]. Similarly, the δ values for the *m*/*z* 191, 179, 135, and 93 fragments were determined as 0.52, 0.56, 2.96, and 5.37 ppm, respectively. The δ value of the molecular ion peak was recorded as 0.58 ppm (Table 1). Similarly, isochlorogenic acid B (**34**), isochlorogenic acid C (**40**), and isochlorogenic acid A (**43**) showed very low δ values (−1.13~5.37 ppm) as well. Therefore, the qualitative analysis of compounds in CoC and CoN can be considered as reliable, and the identification of the compounds is regarded as putative rather than tentative.

### 2.2. Quantitative Analysis by Database-Enabled UHPLC-Q-Orbitrap-MS/MS

In addition to the qualitative analysis, this study also performed a quantitative analysis of CoC and CoN. The main results are shown in Figure 4.

#### 2.2.1. The Content of Sesquiterpenes

Through a comparative analysis of their quantification results, a remarkable difference in sesquiterpenes was found between CoC and CoN as well as between the aerial part of CoC (CoCA) and the underground part of CoC (CoCU). As shown in Figure 4A,B, the sesquiterpene content of CoC was much higher than that of CoN. For example, the sum of atractylenolide III (**54**) in CoC was 613.53 μg/g, while the sum of atractylenolide III (**54**) in CoN was 0.00 μg/g. The total content of four sesquiterpenes (**54**, **56**, **57**, and **58**) in CoC was 980.18 μg/g, while the total content of four sesquiterpenes in CoN was 87.21 μg/g.

This great difference suggests sesquiterpene enrichment as an important feature of CoC. Notably, sesquiterpene atractylenolide III was present in CoC and absent in CoN. Therefore, atractylenolide III could act as the quality marker (Q-marker) to distinguish CoC from CoN. However, it cannot be used to distinguish CoC from *Codonopsis pilosula*, a common traditional Chinese medicine [52]. Furthermore, CoCA and CoCU were also different in sesquiterpene content. As seen in Figure 4A, the sesquiterpene content of CoCA was higher than that of CoCU. This difference might have arisen from the plant’s metabolism. On the other hand, these sesquiterpenes have been proven as bioactive compounds [53,54], and thus differences in their chemical content may also promise different pharmacological advantages of CoC and CoN as well as CoCA and CoCU.

#### 2.2.2. The Content of Quinic Acid Derivatives

Similar to sesquiterpenes, quinic acid derivatives have also changed their contents between the different plant species and even between the different plant parts. As seen in Figure 4C,D, the total content of six quinic acid derivatives of CoC was higher than that of CoN. In both CoC and CoN, the quinic acid derivative contents of the aerial parts were always much higher than those of the corresponding underground parts. This could be used to distinguish the aerial and underground parts of the two Tibetan *Codonopsis* species.

The quinic acid derivatives exhibited versatile pharmacological effects, and thus the difference between the quinic acid derivatives implies that the aerial part and underground part of two Tibetan *Codonopsis* species are not mutually replaceable.

#### 2.2.3. The Content of Flavonoids

In addition to the sesquiterpenes and quinic acid derivatives, flavonoids also varied their contents with the plant species and plant parts. As illustrated in Figure 4E,F, 16 flavonoids were found in this study. In general, the flavonoid content of CoC was higher than that of CoN, while the flavonoid contents of two aerial parts were higher than those of two underground parts, respectively. Two flavonoids, diosmetin (**49**) and chrysoeriol (**51**), were found to mainly occur in the CoCA but were nearly absent in CoCU, the aerial parts of CoN (CoNA), and the underground parts of CoN (CoNU). Thus, the two flavonoids, diosmetin (**49**) and chrysoeriol (**51**), were proposed as the Q-markers of CoCA. The great difference in their distribution also indicates that CoCA cannot be alternatively used as CoCU, CoNA, or CoNU.

#### 2.2.4. The Content of Others

Compared with the differences in sesquiterpenes, quinic acid derivatives, and flavonoids, the differences in other compounds were not so significant. As illustrated in Figure 4G,H, the distributions of amino acid, organic acid, phenol, and lipid compounds were a bit similar to each other between two Tibetan *Codonopsis* species. The distributions of other compounds were basically similar to each other between the two Tibetan *Codonopsis* species as well. Specifically, lobetyolin (**45**) was found to be present in substantial amounts in CoCA, CoCU, CoNA, and CoNU (Figure 4I,J). It is worth mentioning that lobetyolin ubiquitously occurred in *Codonopsis* and possessed significant pharmacological activity [55,56]. Possibly owing to this reason, it was selected as a Q-marker of *Codonopsis pilosula*, a common traditional Chinese medicine [52]. The findings of the ubiquitous presence of lobetyolin further indicate that lobetyolin (**45**) can be used as the common Q-marker of CoCA, CoCU, CoNA, and CoNU.

It is noted that (1), herein, the so-called Q-marker was mainly a chemical Q-marker; it is not involved in pharmacology. (2) Rare and bioactive compounds such as agrimol B (a phloroglucinol derivative, **60**) and eleutheroside E1 (a lignan compound, **37**) were also detected in the study [36,46]. (3) The contents of betaine, sucrose, D-gluconic acid, 1-kestose, and nystose may contribute to the sweetness of CoC and CoN.

### 2.3. Total Polyphenol Content

In addition to differences in the individual compounds, there was also a great difference in the total polyphenol content Appendix A. Total polyphenols were detected to be in higher amounts in the underground parts of both species compared to the aerial parts. This further supports the above findings.

Overall, these findings not only reveal significant differences in the composition of CoC and CoN but also emphasize the distinct distribution of compounds between the aerial and underground parts. These results provide a chemical basis for further studies on these two species. Furthermore, all these differences in individual compounds and total polyphenol support the strategy for preservation of underground parts of two Tibetan *Codonopsis* species. This is because (1) the underground parts cannot be used similarly to the aerial parts, as mentioned above. (2) Preservation of the underground parts and harvesting of the aerial parts can ensure its sustainable development.

## 3. Materials and Methods

### 3.1. Plants Materials and Reagents

CoC and CoN were collected from two locations in the Sichuan Province, China, and their aerial and underground parts were separated (Table 2). All specimens were identified by Professor Guihua Jiang at the Chengdu University of Traditional Chinese Medicine (Chengdu, Sichuan, China). Methanol and water were of mass spectra purity grade. In Appendix A, the information of all authentic standards is listed in detail.

### 3.2. Preparation of the Lyophilized Aqueous Extracts from CoCA, CoCU, CoNA, and CoNU

The lyophilized aqueous extracts of four samples were prepared according to Li’s method [57] to avoid a possible solvent effect. Then, the lyophilized aqueous extracts were dissolved in methanol and filtered through a 0.45 µm membrane to obtain a sample solution of 50 mg/mL [58] (Figure 5).

### 3.3. Preparation of Authentic Standard Solution

All authentic standards were individually dissolved in methanol to get a final concentration of 30 μg/mL. After that, these solutions were separately filtered through a 0.45 μm membrane. All filtrates were then kept in a refrigerator at 2–6 °C for the following analysis.

### 3.4. UHPLC-Q-Orbitrap-MS/MS Conditions

The UHPLC-Q-Orbitrap-MS/MS system (Thermo Fisher Scientific, Waltham, MA, USA) was coupled with an Accucore RP-MS LC C_18_ column (100 mm × 2.1 mm, 2.6 μm, Thermo Fisher, Waltham, MA, USA) for chromatographic separations. Based on a previous study [59], the analytical conditions were listed as follows. The column chamber temperature was kept at 40 °C, and the injection volume was 3 μL. The mobile phase A (0.1% HCOOH) and the mobile phase B (100% CH_3_OH) constituted the binary mobile solvent in the negative model. The mobile phase A (CH_3_COONH_4_ − 0.1% HCOOH mixture) and the mobile phase B (100% CH_3_OH) constituted the binary mobile solvent in the positive model. The gradient elution was set as follows: 0–5 min, 10% B; 5–14.5 min, 10–100% B; 14.5–16 min, 100% B; and 16–16.1min, 100–10% B. The operating parameters were as follows: auxiliary gas, 10; sheath gas, 40; sweep gas, 0; and spray voltage, 4.5 kV. The temperatures of both the auxiliary gas heater and capillary were set at 450 °C. The full MS resolution and dd-MS^2^ were 70,000 and 17,500. Automatic gain control (AGC) target was set at 2 × 10^5^. Nitrogen was applied for spray stabilization and as the damping gas in the C-trap. The stepped normalized collision energy was set as 20, 50, and 100 V [14].

### 3.5. Qualitative Analysis and Putative Identification of Compounds

Data acquisition (retention time, molecular ion peak, and MS/MS spectra) and analysis were performed using Xcalibur 4.1 software package and TraceFinder General Quan (Thermo Fisher Scientific Inc., Waltham, MA, USA) [14]. Putative identification was accomplished by comparison with the database and the manual elucidation of mass spectral fragments to meet Level 1 of the high-resolution mass spectrometry confidence [60].

### 3.6. Quantitative Analysis of Compounds

Authentic standard solutions with a concentration of 0, 1, 2, 10, and 30 μg/mL were prepared in line with Li’s method [61]. These authentic standard solutions of increasing concentrations were injected into the ultra-high performance liquid chromatograph to detect the peak area. The injection volume was 3 μL. The calibration curves were calculated by the peak area and content of the standard solution. According to the calibration curves, the content of each compound (μg/g) in each sample solution was calculated. Each quantitative assessment experiment was repeated three times, and the corresponding standard deviations (SD) were calculated.

### 3.7. Total Polyphenols of CoCA, CoCU, CoNA, and CoNU

The total phenolic content of the extracted compounds was determined using the Folin–Ciocalteu reagent. The absorbance at 760 nm was measured, using 15% Na_2_CO_3_ solution as the blank. The total polyphenol content was expressed in mg/g luteolin equivalents, based on the regression equations obtained from the authentic standard luteolin solution of different concentrations.

### 3.8. Statistical Analysis

Data are given as the mean ± SD of three measurements. The IC_50_ values were calculated by linear regression analysis. All linear regression in this paper was analyzed by the Origin 6.0 professional software. Significant differences were performed using the *t*-test (*p* < 0.05). Analysis was performed using the SPSS software (v.12, SPSS, Chicago, IL, USA).

## 4. Conclusions

A total of 66 compounds, including 16 isomers, were putatively identified and quantified in the aerial and underground parts of CoC and CoN. The distribution of these compounds has been proven to have the following four features: (1) Four sesquiterpenes are enriched mainly in CoC and rarely in CoN. Particularly, sesquiterpene atractylenolide III is present in CoC and absent in CoN, and thus it can act as the Q-marker to distinguish CoC and CoN. (2) Four quinic acid derivatives are abundant in the aerial part of two Tibetan *Codonopsis* species, thus they can characterize the aerial parts of both CoC and CoN. (3) The total content of 16 flavonoids is higher in CoC than in CoN; significantly, two flavonoids, diosmetin and chrysoeriol, are suitable as Q-markers of the aerial part of CoC. (4) Lobetyolin may function as the common Q-marker of both CoC and CoN, for its ubiquitous presence in four parts of both herbs. These findings will help to build a sustainable development strategy and quality control method for these two traditional Tibetan medicinal herbs.

## Figures and Tables

**Figure 1 molecules-29-05203-f001:**
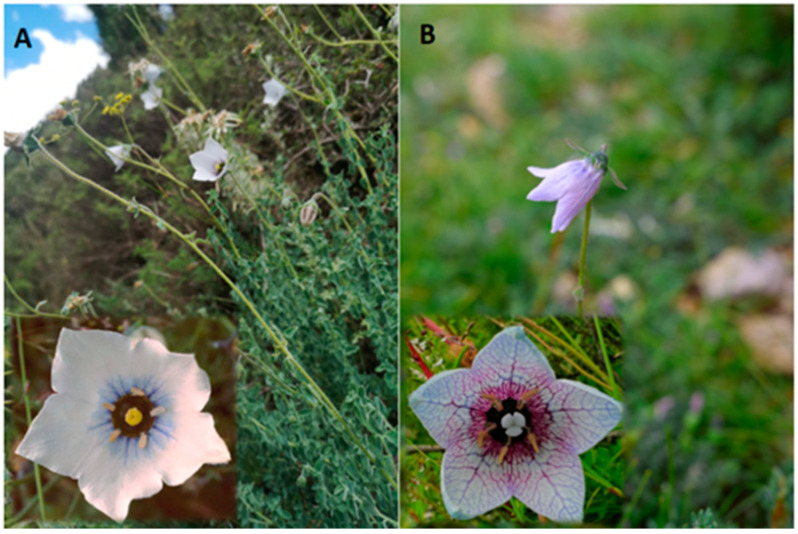
Plant of *Codonopsis canescens* Nannf. (**A**); Plant of *Codonopsis nervosa* Nannf. (**B**).

**Figure 2 molecules-29-05203-f002:**
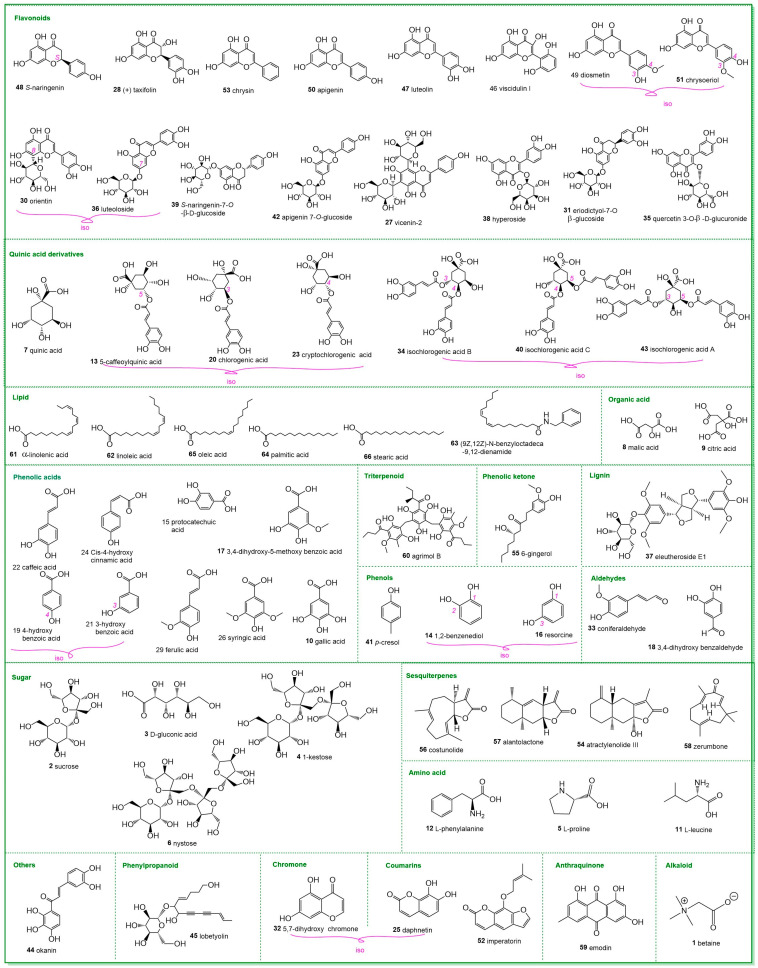
The structures and configurations of the 66 identified compounds (**1**–**66**). Isomers were connected with a knot and denoted by “iso”. The identification process was detailed in Appendix A. The number in purple indicates the numbering of atom in whole molecule.

**Figure 3 molecules-29-05203-f003:**
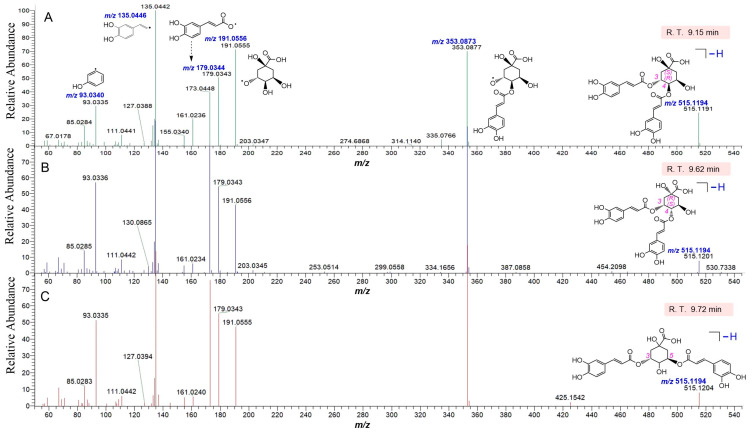
The main evidence for the putative identification of isomers of three quinic acid derivatives (**34**, **40**, and **43**) in two Tibetan *Codonopsis* species. (**A**), MS/MS spectra of isochlorogenic acid B (**34**); (**B**), MS/MS spectra of isochlorogenic acid C (**40**); (**C**), MS/MS spectra of isochlorogenic acid A (**43**). The *m*/*z* values in blue italic are the calculated ones.

**Figure 4 molecules-29-05203-f004:**
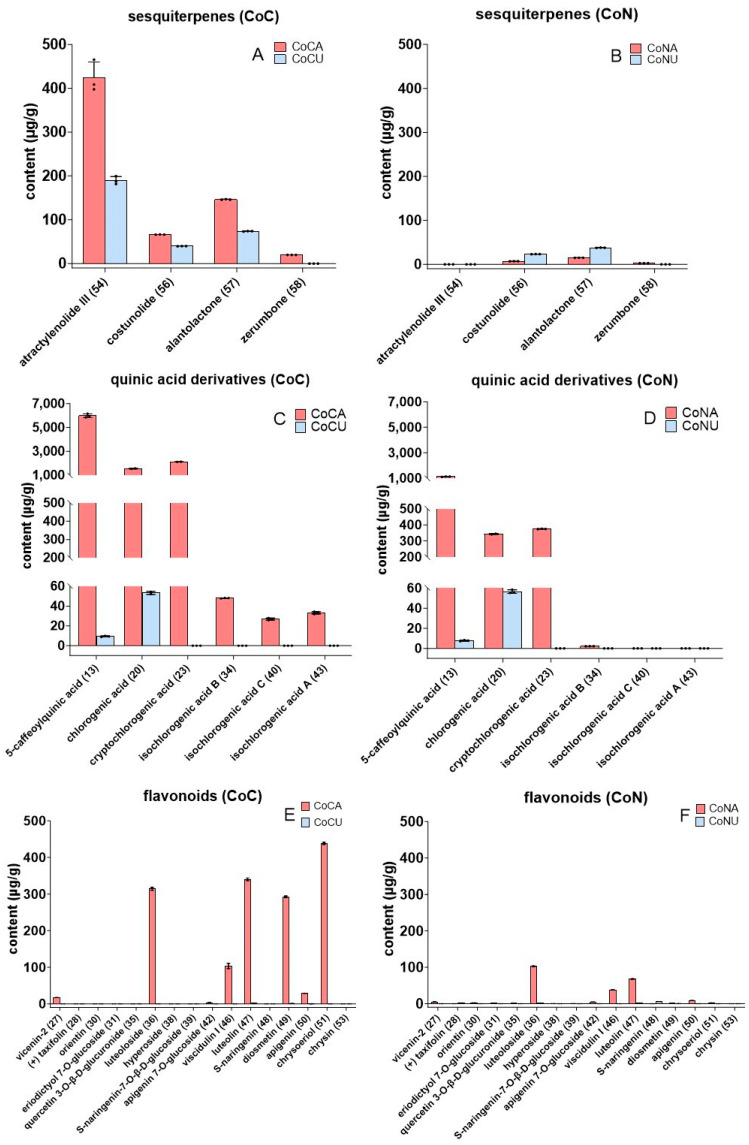
The contents of 66 compounds (**1**–**66**) in CoC and CoN. (**A**) The contents of four sesquiterpenes in CoC. (**B**) The contents of four sesquiterpenes in CoN. (**C**) The contents of 6 quinic acid derivatives in CoC. (**D**) The contents of 6 quinic acid derivatives in CoN. (**E**) The contents of 16 flavonoids in CoC. (**F**) The contents of 16 flavonoids in CoN. (**G**) The contents of amino acid, organic acid, phenol, and lipid in CoC. (**H**) The contents of amino acid, organic acid, phenol, and lipid in CoN. (**I**) The contents of others in CoC. (**J**) The contents of others in CoN. The horizontal lines below the content of 0.0 in the vertical coordinate correspond to value of −10. Each value is expressed as mean ± SD (*n* = 3). The values with letters, regression equation, and the calculation procedure for the quantification of the compounds are also detailed in Appendix A.

**Figure 5 molecules-29-05203-f005:**
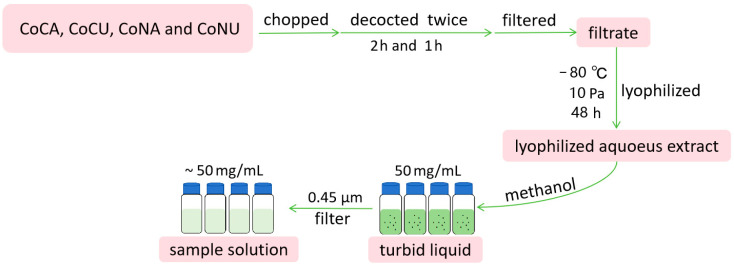
The flowchart for preparing the sample solution of the lyophilized aqueous extracts from CoCA, CoCU, CoNA and CoNU.

**Table 1 molecules-29-05203-t001:** The main experimental results of 66 putatively identified compounds (**1**–**66**).

NO.	Name	R.T. min	Molecular Ion	Observed *m*/*z* Value	Theoretical *m*/*z* Value	Error δ (ppm)	Characteristic MS/MS *m*/*z* Value	Ref.
**1**	Betaine *	0.48	[C_5_H_11_NO_2_ + H] ^+^	118.0865	118.0862	−2.54	100.0761, 72.0814, 58.0659	[12]
**2**	sucrose	0.55	[C_12_H_22_O_11_ − H]^−^	341.1088	341.1089	0.29	179.0554, 101.0234, 71.0128, 59.0127	[13]
**3**	D-gluconic acid	0.57	[C_6_H_12_O_7_ − H]^−^	195.0504	195.0506	1.03	177.0398, 129.0184, 99.0077, 75.0077, 59.0128	[14]
**4**	1-kestose	0.58	[C_18_H_32_O_16_ − H]^−^	503.1617	503.1617	0.00	323.0984, 179.0554, 89.0234, 59.0127	[15]
**5**	*L*-proline *	0.59	[C_5_H_9_NO_2_ − H]^−^	116.0708	116.0706	−1.72	98.0605, 70.0657, 56.0501	[16]
**6**	nystose	0.6	[C_24_H_42_O_21_ − H]^−^	665.2144	665.2145	0.15	485.1509, 341.1087, 113.0234, 71.0127	[17]
**7**	quinic acid	0.62	[C_7_H_12_O_6_ − H]^−^	191.0555	191.0561	3.14	173.0446, 127.0391, 109.0286, 85.0285, 59.0128	[18]
**8**	malic acid	0.63	[C_4_H_6_O_5_ − H]^−^	133.0134	133.0142	6.01	115.0028, 89.0234, 71.0128	[19]
**9**	citric acid	0.8	[C_6_H_8_O_7_ − H]^−^	191.0192	191.0197	2.62	173.0086, 129.0184, 111.0078, 87.0077, 57.0335	[20]
**10**	gallic acid	1.01	[C_7_H_6_O_5_ − H]^−^	169.0135	169.0142	4.14	125.0235, 107.0129, 97.0284, 69.0335	[21]
**11**	*L*-leucine	1.02	[C_6_H_13_NO_2_ − H]^−^	130.0864	130.0873	6.92	102.9875, 88.0392, 69.0335	[22]
**12**	*L*-phenylalanine	1.56	[C_9_H_11_NO_2_ − H]^−^	164.0709	164.0717	4.88	147.0443, 94.0286, 72.0080	[20]
**13**	5-caffeoylquinic acid	1.72	[C_16_H_18_O_9_ − H]^−^	353.0877	353.0878	0.28	191.0555, 161.0237, 135.0443, 85.0284, 59.0128	[23]
**14**	1,2-benzenediol	1.83	[C_6_H_6_O_2_ − H]^−^	109.0285	109.0295	9.17	108.0207, 91.0179, 81.0334, 65.0021	[24]
**15**	protocatechuic acid	1.85	[C_7_H_6_O_4_ − H]^−^	153.0185	153.0193	5.23	109.0285, 91.0179, 81.0335, 65.0021	[20]
**16**	resorcine	2.03	[C_6_H_6_O_2_ − H]^−^	109.0285	109.0295	9.17	108.0207, 91.0179, 81.0334	[25]
**17**	3,4-dihydroxy-5-methoxybenzoic acid	2.79	[C_8_H_8_O_5_ − H]^−^	183.0291	183.0298	3.82	168.0057, 139.0392, 124.0157, 95.0125	[14]
**18**	3,4-dihydroxybenzaldehyde	2.94	[C_7_H_6_O_3_ − H]^−^	137.0235	137.0244	6.57	136.0157, 108.0207, 95.0127, 81.0335	[26]
**19**	4-hydroxybenzoic acid	3.64	[C_7_H_6_O_3_ − H]^−^	137.0235	137.0244	6.57	93.0335, 75.0228, 65.0385	[27]
**20**	chlorogenic acid	3.87	[C_16_H_18_O_9_ − H]^−^	353.0883	353.0878	−1.42	191.0555, 127.0391, 85.0284, 59.0127	[23]
**21**	3-hydroxybenzoic acid	3.99	[C_7_H_6_O_3_ − H]^−^	137.0235	137.0244	6.57	108.0208, 93.0335, 65.0384	[28]
**22**	caffeic acid	4.55	[C_9_H_8_O_4_ − H]^−^	179.0343	179.0349	3.35	135.0443, 117.0335, 90.9969	[12]
**23**	cryptochlorogenic acid	5.77	[C_16_H_18_O_9_ − H]^−^	353.0879	353.0878	−0.28	191.0555, 173.0448, 135.0443, 93.0336, 67.0179	[23]
**24**	*cis*-4-hydroxycinnamic acid	6.59	[C_9_H_8_O_3_ − H]^−^	163.0393	163.0400	4.29	119.0493, 104.0253, 93.0336	[29]
**25**	daphnetin	7.22	[C_9_H_6_O_4_ − H]^−^	177.0186	177.0193	3.95	149.0235, 121.0286, 93.0336, 65.0385	[30]
**26**	syringic acid	7.3	[C_9_H_10_O_5_ − H]^−^	197.0448	197.0455	3.55	182.0214, 153.0551, 123.0079, 95.0129	[12]
**27**	vicenin-2	8.22	[C_27_H_30_O_15_ − H]^−^	593.1511	593.1511	0.00	473.1098, 353.0663, 297.0768, 117.0336, 93.0333	[31]
**28**	(+) taxifolin	8.53	[C_15_H_12_O_7_ − H]^−^	303.0512	303.0510	−0.66	285.0404, 199.0399, 125.0235, 57.0335	[32]
**29**	ferulic acid	8.66	[C_10_H_10_O_4_ − H]^−^	193.0500	193.0506	3.11	178.0265, 149.0601, 134.0365, 106.0416	[12]
**30**	orientin	8.68	[C_21_H_20_O_11_ − H]^−^	447.0933	447.0932	−0.22	357.0612, 327.0510, 299.0560, 227.0339, 133.0286	[33]
**31**	eriodictyol 7-*O*-glucoside	8.83	[C_21_H_22_O_11_ − H]^−^	449.1110	449.1089	−4.68	287.0561, 151.0029, 135.0443, 107.0128, 65.0022	[24]
**32**	5,7-dihydroxychromone	8.89	[C_9_H_6_O_4_ − H]^−^	177.0186	177.0193	3.95	149.0236, 133.0286, 91.0179, 63.0229	[34]
**33**	coniferaldehyde	9.02	[C_10_H_10_O_3_ − H]^−^	177.0549	177.0557	4.52	162.0314, 134.0367, 93.0332	[35]
**34**	isochlorogenic acid B	9.15	[C_25_H_24_O_12_ − H]^−^	515.1191	515.1194	0.58	353.0877, 191.0555,179.0343, 135.0442, 93.0335	[23]
**35**	quercetin 3-*O*-β-D-glucuronide	9.25	[C_21_H_18_O_13_ − H]^−^	477.0669	477.0674	1.05	301.0352, 227.0349, 151.0029, 109.0284, 65.0022	[24]
**36**	luteoloside	9.27	[C_21_H_20_O_11_ − H]^−^	447.0933	447.0932	−0.22	285.0403, 199.0396, 133.0286, 107.0129, 63.0229	[24]
**37**	eleutheroside E1	9.33	[C_28_H_36_O_13_ − H]^−^	579.2085	579.2083	−0.35	417.1555, 387.1086, 181.0500, 151.0029, 123.0078	[36]
**38**	hyperoside	9.36	[C_21_H_20_O_12_ − H]^−^	463.0876	463.0881	1.08	300.0273, 271.0248, 243.0297, 199.0390, 151.0029	[20]
**39**	S-naringenin-7-*O*-β-D-glucoside	9.43	[C_21_H_22_O_10_ − H]^−^	433.1133	433.1140	1.62	271.0609, 151.0030, 119.0493, 93.0336	[24]
**40**	isochlorogenic acid C	9.62	[C_25_H_24_O_12_ − H]^−^	515.1201	515.1194	−1.36	353.0877, 191.0556, 179.0343, 135.0443, 93.0336	[23]
**41**	*p*-cresol *	9.64	[C_7_H_8_O − H]^−^	109.0651	109.0647	−3.67	107.0493, 94.0418, 79.0548, 65.0392	[37]
**42**	apigenin 7-*O*-glucoside	9.71	[C_21_H_20_O_10_ − H]^−^	431.0981	431.0983	0.46	268.0377, 211.0395, 117.0337, 63.0229	[38]
**43**	isochlorogenic acid A	9.72	[C_25_H_24_O_12_ − H]^−^	515.1204	515.1194	−1.94	353.0877, 191.0555,179.0343, 135.0443, 93.0335	[23]
**44**	okanin	10.13	[C_15_H_12_O_6_ − H]^−^	287.0562	287.0561	−0.35	269.0467, 151.0028, 135.0442	[39]
**45**	lobetyolin *	10.13	[C_20_H_28_0_8_ + NH4]^+^	414.2119	414.2119	0.00	217.1225, 199.1117, 128.0622, 93.0340	[38]
**46**	viscidulin I	10.47	[C_15_H_10_O_7_ − H]^−^	301.0352	301.0353	0.33	245.0446, 178.9979, 151.0028, 107.0129, 65.0022	[14]
**47**	luteolin	10.72	[C_15_H_10_O_6_ − H]^−^	285.0403	285.0404	0.35	241.0501, 175.0394, 151.0028, 133.0286, 107.0128, 65.0022	[24]
**48**	S-naringenin	10.74	[C_15_H_12_O_5_ − H]^−^	271.0611	271.0611	0.00	187.0397, 151.0028, 119.0493, 83.0128, 65.0022	[14]
**49**	diosmetin	11.23	[C_16_H_12_O_6_ − H]^−^	299.0560	299.0561	0.33	284.0325, 256.0375, 183.0442, 133.0285, 83.0128	[20]
**50**	apigenin	11.25	[C_15_H_10_O_5_ − H]^−^	269.0454	269.0455	0.37	225.0550, 151.0029, 117.0336, 83.0128, 65.0022	[20]
**51**	chrysoeriol	11.34	[C_16_H_12_O_6_ − H]^−^	299.0560	299.0561	0.33	284.0326, 256.0375, 227.0343, 107.0126, 65.0020	[40]
**52**	imperatorin *	12.21	[C_16_H_14_O_4_ + H]^+^	271.0961	271.0964	1.11	203.0339, 147.0442, 91.0549, 69.0705	[41]
**53**	chrysin	12.27	[C_15_H_10_O_4_ − H]^−^	253.0505	253.0506	0.40	209.0603, 181.0641, 143.0495, 107.0127, 63.0229	[42]
**54**	atractylenolide III	12.49	[C_15_H_20_O_3_ − H]^−^	247.1338	247.1339	0.40	203.1436, 187.1122, 135.0806, 83.0491	[38]
**55**	6-gingerol	12.55	[C_17_H_26_O_4_ − H]^−^	293.1761	293.1758	−1.02	236.1052, 221.1543, 177.0918, 148.0521, 71.0127	[35]
**56**	costunolide *	12.77	[C_15_H_20_O_2_ + H]^+^	233.1534	233.1536	0.86	187.1481, 131.0856, 91.0547, 67.0549	[43]
**57**	alantolactone *	12.87	[C_15_H_20_O_2_ + H]^+^	233.1533	233.1536	1.29	187.1479, 131.0856, 91.0547, 67.0549	[44]
**58**	zerumbone *	13.58	[C_15_H_22_O + H]^+^	219.1744	219.1743	−0.46	145.1011, 119.0857, 91.0547, 67.0549	[45]
**59**	emodin	13.63	[C_15_H_10_O_5_ −H]−	269.0455	269.0455	0.00	225.0554, 182.0369, 157.0653	[38]
**60**	agrimol B	14.7	[C_37_H_46_O_12_ − H]^−^	681.2905	681.2916	1.61	308.4896, 187.9433 78.9581	[46]
**61**	α-linolenic acid	14.85	[C_18_H_30_O_2_ − H]^−^	277.2171	277.2173	0.72	207.7533, 99.5756, 71.0129	[47]
**62**	linoleic acid	15.1	[C_18_H_32_O_2_ − H]^−^	279.2328	279.2329	0.36	173.9629, 91.6509, 71.0874	[38]
**63**	(9Z, 12Z)-N-benzyloctadeca-9, 12-dienamide *	15.13	[C_25_H_39_NO + H]^+^	370.3097	370.3104	1.89	108.0812, 91.0547, 65.0393	[48]
**64**	palmitic acid	15.26	[C_16_H_32_O_2_ − H]^−^	255.2328	255.2329	0.39	219.4489, 121.5311, 91.6502	[49]
**65**	oleic acid	15.35	[C_18_H_34_O_2_ − H]^−^	281.2484	281.2486	0.71	149.8671, 123.4038, 79.9083	[50]
**66**	stearic acid	15.59	[C_18_H_36_O_2_ − H]^−^	283.2643	283.2642	−0.35	180.6442, 140.6968, 94.3611	[51]

Note: Compounds with the “*” sign were identified in positive ion mode. Despite the scan mode range being set at *m*/*z* 100–1200 in the mass spectra, *m*/*z* values < 100 were also detected by the Xcalibur 4.1 Software package. The *m*/*z* values of characteristic MS/MS were adopted from the sample solution rather than authentic standard solution. The original MS spectra and identification process were detailed in Appendix A. The total ion chromatogram (TIC) of CoC and CoN are shown in Appendix A.

**Table 2 molecules-29-05203-t002:** The information of collection of CoC and CoN.

Samples	Part	Sources	Altitude	Collection Date
CoC	Aerial parts	Luhuo, Sichuan, China	3257.3 m	2023.8
CoC	Underground parts	Luhuo, Sichuan, China	3257.3 m	2023.8
CoN	Aerial parts	Ma’erkang, Sichuan, China	3940 m	2023.8
CoN	Underground parts	Ma’erkang, Sichuan, China	3940 m	2023.8

## Data Availability

The data presented in this study are available on request from the corresponding author.

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
