# Peer review of "Simultaneous Quantification of 66 Compounds in Two Tibetan *Codonopsis* Species Reveals Four Chemical Features by Database-Enabled UHPLC-Q-Orbitrap-MS/MS Analysis"

_molecules, 2024, doi:10.3390/molecules29215203_

Round 1

Reviewer 1 Report

Comments and Suggestions for Authors

The authors present the quantification of 66 compounds in two Tibetan codonopsis species  by suspect  screening MS/MS Analysis. The two species seem to have a special interest to TCM, therefore their thorough analysis is of merit for the readers, especially those interestd in TCM. The authors have made an impresive effort collecting these substances and using them as standards, therefore their results are unequivocal. The methods and the are clearly presented and the organization of the manuscript is adequate. Nevertheless there are some point that deserve clarification or improvement.

At first the authors have used these substances for the comprehensive characterization of the species, but they should clarify the level of identification obtained. The paper by Szymanski (DOI: 10.1021/es5002105) provides a robust methodology to do so. On identifying the substances, the authors should also provide the algorithmic means to do so is its e.e. a cosine similarity of MSMS spectra that enabled the identification, and what are the cutoff levels?

The authors also provided evidence of differences between the species. This is interesting but it should some statistical data should also be provided, given that they analyzed three samples. At least a t-test for every substance differing should provide the reliability of these observations.

The calibration curves provided are not in a good shape as there are not formulated with the correct number of significant digits nor they provide the std error of the slope and the intercept at least. There is also an alarming point. Theauthors have stated that the range of selected calibration curves icludes 0. This should be stated. If the 0.0 point (which is the theoretical blank) is icluded in the calibration curve or did the curve pass through this point by coersion? An the authors should also answer why thehavent used this methodology for all calibration curves. Therefore I would suggest that the no of points for each calibration curve should be stated clearly.  A final comment. The 0.00 point stated in the abstract as the lowest concentration is not correct. The authors did not presumably measured 0.00 concentration so this should be modified refering the lowest point they actually included to their calibration curve.

Reviewer 2 Report

Comments and Suggestions for Authors (1) The article uses the UHPLC-Q-Orbitrap-MS/MS method for qualitative analysis to determine the types of substances present and whether there is evidence to support that the substances present are the ones claimed rather than others.

(2) In section 2.1, why were [M - H]-, [M + H]+, and [M + NH4]+ used as analytical models?

(3) Preparation method is very important. What were the advantages of the chosen lyophilized aqueous extracts preparation method?

(4) How is the linearity range determined for all compounds in the Suppl. 3 Semi‑Quantification of compounds?How is the limit of quantification determined?Why 1.0? Why 0?

(5) In the Suppl. 3 Semi‑Quantification of compoundsIf R the value is too small, you are advised to delete it! Such as 0.510

(6) Significant numbers in the table are inconsistent!
